# Cytological and Molecular Characterization of a New Ogura Cytoplasmic Male Sterility Restorer of *Brassica napus* L.

**DOI:** 10.3390/plants13121703

**Published:** 2024-06-19

**Authors:** Lan Huang, Yun Ren, Baogang Lin, Pengfei Hao, Kaige Yi, Xi Li, Shuijin Hua

**Affiliations:** 1Institute of Crop and Nuclear Technology Utilization, Zhejiang Academy of Agricultural Sciences, Hangzhou 310021, China; huang975885829@163.com (L.H.); linbg@mail.zaas.ac.cn (B.L.); 11816004@zju.edu.cn (P.H.); kaige_y0113@163.com (K.Y.); 202200101021@stu.xza.edu.cn (X.L.); 2College of Advanced Agricultural Sciences, Zhejiang A & F University, Hangzhou 311300, China; 3Institute of Crop, Huzhou Academy of Agricultural Sciences, Huzhou 313000, China; yunhuaren@163.com

**Keywords:** *Brassica napus*, cytology, molecular marker, Ogura CMS, restorer line

## Abstract

Ogura cytoplasmic male sterility (CMS) is considered the rapeseed (*Brassica napus* L.) with the most potential to be utilized as a heterosis system worldwide, but it lacks sufficient restorers. In this study, root tip cell (RTC) mitotic and pollen mother cell (PMC) meiosis observations were compared to ensure the number of chromosomes and the formation of a chromosomal bridge using restorer lines R2000, CLR650, and Zhehuhong (a new restorer) as the experimental material. Further, molecular markers of exogenous chromosomal fragments were detected and the sequence and expression differences of restorer genes in the three lines were determined to identify the distinctive characteristics of Zhehuhong. The results showed that the number of chromosomes in Zhehuhong was stable (2n = 38), indicating that the exogenous radish chromosome segment had been integrated into the chromosome of Zhehuhong. Molecular marker detection revealed that Zhehuhong was detected at most loci, with only the RMA05 locus being missed. The exogenous radish chromosome segment of Zhehuhong differed from R2000 and CLR650. The pollen mother cells of Zhehuhong showed chromosome lagging in the meiotic metaphase I, meiotic anaphase I, and meiotic anaphase II, which was consistent with R2000 and CLR650. The restorer gene *PPRB* in Zhehuhong had 85 SNPs compared with R2000 and 119 SNPs compared with CLR650, indicating the distinctive characteristic of *PPRB* in Zhehuhong. In terms of the spatial expression of *PPRB*, the highest level was detected in the anthers in the three restorer lines. In addition, in terms of temporal expression, the *PPRB* gene expression of Zhehuhong was highest at a bud length of 4 mm. Our results clearly indicated that Zhehuhong is a new restorer line for the Ogura CMS system, which can be used further in rapeseed heterosis utilization.

## 1. Introduction

Rapeseed is an important crop for edible oil. Increasing seed yield is an essential goal in rapeseed production. Both genetic improvement and suitable agronomic practices are vital for obtaining a high seed yield in rapeseed. On the manipulation of genetic improvement, heterosis utilization is effective in enhancing rapeseed yield [1,2]. Among heterosis utilization systems, the cytoplasmic male sterility (CMS) system is the major type, which mainly includes Polima CMS and Ogura CMS [3,4]. Currently, more than 80% of rapeseed varieties belong to hybrids including Polima CMS, recessive genic male sterility, and chemical hybridization in China. However, the identification of Polima CMS is a milestone for rapeseed hybrid application in China [5,6,7]. The iconic national rapeseed variety ‘Qinyou 2’ was a product based on the Polima CMS system, released in 1992 [8]. The area for growth in this variety was more than three hundred million hectares [8]. The yield contribution of rapeseed hybrids increased from 1950.0 to 2689.8 kg ha^−1^ until 2014 [9], indicating the importance of the application of rapeseed CMS in production.

In addition to Polima CMS, the Ogura CMS system of rapeseed (*Brassica napus*) is another important pathway for rapeseed heterosis utilization. This system has been widely applied in rapeseed production in many countries, such as France and Canada [10,11]. One of the advantages for the Ogura CMS sterile line is its complete microspore or pollen abortion, which shows stable sterility during climate changes [12,13,14,15]. However, the Ogura CMS system suffers from a deficiency in the restorer genes in rapeseed, which normally originated from radish (*Raphanus sativus* L.) [16,17,18,19,20]. Therefore, the successful breeding of the restorer line is key to achieving the Ogura CMS hybridization system in rapeseed production.

Although the breeding of Ogura CMS restorer lines has already been carried out, progress has been relatively slow. Ogura CMS was discovered by Ogura in *Raphanus sativus*, a relative of *Brassica napus*, in Kagoshima, Japan [21]. Heyn and Sakai obtained the new restorer lines through sexual and asexual hybridization, but due to partial fertility, they could not be directly used in rapeseed production [22,23]. In the 21st century, the breeding of the Ogura CMS restorer line made a breakthrough. The Institut National de la Recherche Agronomique (INRA) bred a new variety with a double low (low erucic acid and glucosinolate content) restorer line R2000 through gamma ray radiation mutagenesis technology and consecutive generations of backcross breeding [10]. The restorer line performed well in agronomic traits, seed setting rate, and fertility restoration. For example, the fertile/sterile ratio matched 3:1 in self-pollination in generations F_3_ and F_4_ in R2000. In generation F_5_, the number of seeds per pod was significantly higher in R2000 than in R211 (the number of seeds per pod in the R2000 F_5_ in self-pollination and in test-crossing was 26.5 and 27.0, respectively, while that in R211 was 11.2 and 25.5, respectively) [10]. Therefore, it was adopted by growers in Europe, America, Australia, and other countries. However, because the restorer line (R2000) and the corresponding sterile line for the hybrid system are protected by international patents, the use of R2000 in other organizations comes with the payment of expensive patent royalties. By contrast, the research on the Ogura CMS restorer line creation in China has been slow, with only a few reports. Chen et al. (2013) successfully introduced the radish restorer gene into sterile line H13A through grafting technology [24]. Then, they developed a new restorer line of Ogura CMS of rapeseed, CLR650. As the first Ogura CMS restorer line with independent intellectual property rights in China, the breeding of CLR650 enriched the restorer sources for the Ogura CMS system. However, the restorer CLR650 had some defects, including unstable chromosome numbers (2n = 38–40), a low seed setting after self-pollination (0.6–17.4%), and a high seed glucosinolate content (about 80 μmol·g^−1^) [24]. Consequently, it was difficult to directly apply to the commercial production of Ogura CMS hybrid rapeseed. Subsequently, Wang et al. (2020) used CLR650 as the material to breed the new restorer line CLR6430 through crossing with double low line 20B, with subsequent backcrossing and selfing of the F1 hybrids [25]. Compared to CLR650, the content of glucosinolate in CLR6430 reduced by 3.3%, and the number of siliques per plant and the number of seeds per silique of CLR6430 increased by 72.88% and 43.42%, respectively. However, due to the relatively long length of the introduced radish fragments as well, the content of glucosinolates of CLR6430 was still higher than the current standard in China (<40 μmol·g^−1^) [26]. Therefore, further improvements and the creation of new pollen fertility restorers for the Ogura CMS are still needed.

It was reported that there were a large number of Ogura CMS restorer genes in radish [27,28,29,30]. Unfortunately, this restorer is currently widely applied only in R2000 and its derivatives. As a major producer and consumer of rapeseed oil, it was particularly urgent for China to breed new rapeseed Ogura CMS restorers in production [31]. Currently, our research team has successfully bred an Ogura CMS restorer line, Zhehuhong, with red petals [32]. When we conducted a cross between the Ogura CMS sterile line and Zhehuhong, the fertility of F_1_ was 100%. Therefore, we preliminarily hypothesized that Zhehuhong is a new Ogura CMS restorer. However, it was only proven that Zhehuhong is a potential Ogura CMS restorer, so more cytological and molecular evidence is needed. Furthermore, it is not known whether Zhehuhong is different from existing restorers, such as R2000 and CLR650. Given that they are different restorer lines, what are the differences among them? To answer this, the current investigation aimed to characterize whether Zhehuhong is a new restorer line of Ogura CMS, and it also aimed to characterize the differences among the three restorers from mitosis and meiosis analyses, the exogenous chromosome fragment length, the restorer gene sequence (*PPRB*), and its expression profile. This study would therefore provide cytological and molecular evidence for a newly created rapeseed Ogura CMS restorer line, enhancing heterosis utilization.

## 2. Results

### 2.1. Chromosome Number of Ogura CMS Restorer Line of Zhehuhong

In the observation of mitosis in the root tip cells of three Ogura CMS restorer line seedlings, R2000, CLR650, and Zhehuhong, it was found that the number of chromosomes in the root tip cells of Zhehuhong was 2n = 38. This was consistent with the number of chromosomes in *Brassica napus* and suggests that the number of somatic chromosomes in Zhehuhong was stable (Figure 1).

### 2.2. Meiosis Observation of Ogura CMS Restorer Line Zhehuhong

Meiosis analysis was conducted to further compare the effect of the introgressed radish fragment on the rapeseed chromosomal behavior in R2000, CLR650, and Zhehuhong. Individual chromosome lagging was observed in all restorers. Specifically, in R2000, this occurred in meiotic metaphase I and meiotic anaphase II (Figure 2a-1,f-1). In CLR650, this occurred in meiotic metaphase I, meiotic anaphase I, and meiotic anaphase II (Figure 2a-2,b-2,f-2). In Zhehuhong, this occurred in meiotic metaphase I, meiotic anaphase I, and meiotic anaphase II (Figure 2a-3,b-3,f-3). Taken together, chromosomal lagging is a common characteristic of the exogenous radish chromosome fragment in rapeseed during meiosis in the restorer line.

### 2.3. The Length of Exogenous Radish Fragment of Ogura CMS Restorer Zhehuhong

The 79 selected molecular markers were used for detection of the length of the introgressed radish fragment in three restorer lines: R2000, CLR650, and Zhehuhong. There were 76, 78, and 78 markers identified in these lines, respectively (Appendix A). In R2000, three markers, E32M59A, E32M59B, and OPH03, in the fourth group were not detected. The OPH03 marker in the fourth group and RMA05 in the first group were not detected in CLR650 and Zhehuhong, respectively (Figure 3). This clearly indicated that the exogenous radish fragments introduced into CLR650 and Zhehuhong were much longer than those of R2000.

### 2.4. Sequence Characteristics of the Restorer Gene PPRB in Zhehuhong

In order to further identify the differences in the sequences of the restorer gene, we chose *PPRB* for sequencing because *PPRA* has no function in fertility restoration and *PPRC* is a pseudogene [34,35]. It was shown that there were 81 SNPs and three indels in CLR650, compared to R2000, with a homology of 96.16%. In addition, there were 85 SNPs and 15 indels in Zhehuhong compared to R2000, with a homology of 95.43%. In contrast to CLR650 and R2000, there were 119 SNPs and 11 indels, with a homology of 94.05% (Figure 4). Conclusively, there were significant differences in the nucleotide sequences of the *PPRB* gene among the restorer lines R2000, CLR650, and Zhehuhong.

### 2.5. The Spatiotemporal Expression Characteristics of the PPRB in Ogura CMS Restorer Lines

In order to further explore the spatial expression differences of the *PPRB* gene in the three restorer lines, qRT-PCR technology was used to detect the relative expression levels of this gene in the different organs of R2000, CLR650, and Zhehuhong, using the root as the control. It was shown that the level in R2000 was highest in the anthers, followed by the bracts, and lowest in the roots. Furthermore, in the anther it was 28.92 times higher than that in the roots. In CLR650, it was highest in the anther, followed by the bract and stigma, and lowest in the leaf. The level in the anthers was 74.94 times higher than that in the leaves. The *PPRB* in the anther of Zhehuhong was still highest, followed by the stigma, and lowest in the stem. The level in the anther was 2649.13 times higher than that in the stem. Furthermore, among the anthers of the three restorer lines, the *PPRB* gene was highest in CLR650, followed by Zhehuhong, and lowest in R2000. Those in CLR650 and Zhehuhong were 29.20 and 22.12 times higher than that in R2000, respectively (Figure 5a).

In terms of the differences in the temporal expression levels of *PPRB* in different restorer lines, the level in R2000 was generally lower than in CLR650 and Zhehuhong when the bud grew from 2 to 7 mm, reaching its peak at 4 mm and its lowest at 3 mm. The level at 4 mm was 4.19 times higher than at 3 mm. In CLR650, it was highest at 2 mm and lowest at 7 mm. The level at 2 mm was 3.44 times higher than that at 7 mm. In Zhehuhong, it was highest at 4 mm and lowest at 7 mm. The level at 4 mm was 7.58 times higher than that of 7 mm (Figure 5b).

In summary, in terms of spatial expression levels, the *PPRB* of R2000, CLR650, and Zhehuhong was at the highest levels in the anthers. In terms of temporal expression, the *PPRB* gene of R2000, CLR650, and Zhehuhong had the highest levels in the bud, with lengths of 4, 2, and 4 mm, respectively.

## 3. Discussion

Ogura CMS is one of the nucleo-cytoplasmic male sterility types to utilize the heterosis of rapeseed [10,36,37]. It has been widely used in production because of its advantages of complete pollen abortion and easy maintenance [11]. However, the restorer genes of this system only exist in radish, so the difficulty of this system is the creation of restorer lines. In the early stage, our research team successfully introduced a radish fragment with a restorer gene into rapeseed through distant hybridization and microspore culture techniques. Then, a new restorer line, Zhehuhong, was developed [32]. The aim of this study was to identify whether the newly developed restorer line Zhehuhong was a new Ogura CMS restorer via validation using cytological and molecular biological methods. At the same time, it would lay a foundation for expanding China’s new restorer sources for Ogura CMS heterosis utilization systems.

### 3.1. Cytological Characteristics of New Ogura CMS Restorer Line Zhehuhong

The present cytological result showed that the chromosome number of Zhehuhong was 2n = 38, which was consistent with the chromosome number of *Brassica napus* [38]. This indicates that the exogenous radish fragment had been successfully integrated into the genome of *Brassica napus* during distant hybridization between radish and rapeseed. Compared with previous studies, the chromosome number of Zhehuhong was different from those of CLR650 (2n = 38–40) and R2000 (2n = 40) [10,24]. This difference might be due to them having different donors during distant hybridization. Both the restorer gene donors of Zhehuhong and R2000 are from the radish line, but one is Chinese radish (Yidianhong, a local line), while another is European radish (D81) [28]. This means that they have different genetic backgrounds. In CLR650, the restorer gene donor is from *Raphanobrassica* (AACCRR, 2n = 58) [39]. In the distant hybridization of crop species, different donors can result in more hybrids with different allele combinations, which is an effective method for producing new germplasm [40,41]. Meanwhile, the chromosome lagging phenomenon appeared in the three restorers, and this is likely to be related to the length of the introduced radish fragments and the incompatibility of chromosomes [42]. Although chromosome lagging during meiosis did not affect the normal formation of tetrads and pollen grains, it may lead to segregation distortion in the ratio between the fertile and sterile offspring of rapeseed. This phenomenon had also been preliminarily reported in related studies of CLR650 and CLR6430 [15,17]. In CLR650, the percentages of self- and backcross-fertile plants were mostly 50% and 30%, respectively [17]. In CLR6430, they were about 35% and 20%, respectively [26]. In order to reveal the specific location and characteristics of the radish fragment in Zhehuhong, subsequent research could use genomic in situ hybridization (GISH) and bacterial artificial chromosome fluorescence in situ hybridization (BAC-FISH) technology for physical localization.

### 3.2. Molecular Characteristics of New Ogura CMS Restorer Zhehuhong

The molecular marker detection results showed that the length of the radish fragments introduced in Zhehuhong might be similar to that of CLR650, but longer than that of R2000. It is difficult to conclude whether the exogenous radish fragment in Zhehuhong was longer than that in CLR650, although both lost one marker. Their different locations for the lost marker suggested that the donor of exogenous radish fragments was different. Another possible reason for the inability to compare their fragment size was the differing introgression methods for Zhehuhong and CLR650. The introgression of radish fragments to the rapeseed chromosome in Zhehuhong was performed using the distant hybridization method, while grafting was used in CLR650 [24,32]. During distant hybridization, the random exchange of chromosomal fragments takes place between radish and rapeseed, and a stable condition will ultimately be formed after several generations of self-pollination. For the grafting method, the first generation of plants maintains its own sets of chromosomes. However, during self-pollination in the next generation, the exchange of chromosomes between two species will occur and most of the exogenous chromosome fragments will be lost. The process of chromosomal exchange had some common behaviors, such as the randomness and substantial separation of many agronomic traits after the first generation [43,44,45]. In the current study, we did not map the genetic distance of exogenous radish fragments in Zhehuhong. However, a considerable investigation of the radish fragments in R2000 and the new developed lines from CLR65, such as CLR6430, had been conducted. It was reported that the radish fragment in CLR6430 was 72.4 cM [28,46,47]. However, the exact length of exogenous radish chromosome fragments is unclear in all three lines. Other researchers reported that the restorer gene from different sources, including *R. raphanistrum* from Turkey, the Japanese wild radish, European cultivars, and lines from China, resulted in the same or similar loci [48,49]. However, the length of the exogenous radish chromosome was expected to be shorter than that of R2000, because R2000 was bred using irradiation to reduce the length of exogenous radish chromosome fragments [10]. As is known, subjecting organisms to X-ray irradiation can lead to chromosome fragment breakage [50]. As proven, the exogenous radish chromosome fragment can be efficiently shortened, although other methods have also been established to achieve this. The different lengths of exogenous radish fragments between Zhehuhong and R2000 indicated that they are different Ogura CMS restorer lines.

Another important source of evidence for the differences among R2000, CLR650, and Zhehuhong was the different sequences of the restorer gene, *PPRB*. Previous reports showed that there are three alleles of this gene in the Ogura CMS restorer line: *PPRA*, *PPRB*, and *PPRC* [28,34]. However, *PPRA* has no function in fertility restoration and *PPRC* is a pseudogene [34,35]. Only *PPRB* has a function in fertility restoration in the rapeseed Ogura CMS system [51]. Therefore, we only cloned *PPRB* for their sequence comparisons. Although they had a very high sequence similarity, there were clear differences in their sequences among the three restorer lines. Among those differences, it can be found that *PPRB* had many SNPs up- and down-stream. Three indels were also identified. There was a large indel with 41 base pairs in Zhehuhong from position 1258, which clearly differed from those of R2000 and CLR650. Inserts of 12 and six base pairs from position 1654 of Zhehuhong and 2074 of CLR650 differed from those of R2000, respectively. Different frequencies of the base alteration and fragment deletion/insertion indicated the different sources of the restorer gene in the three Ogura restorer lines. However, the reasons for their preference are still unknown and require more experimental evidence.

### 3.3. The Spatiotemporal Expression Differences of PPRB among Three Ogura CMS Restorers

In order to further evaluate the difference in *PPRB* in the three Ogura CMS restorer lines, a spatiotemporal expression analysis by qRT-PCR was conducted. It was shown that higher expression levels were found in the anther in the three lines, which indicated that the restorer gene *PPRB* mainly played its role in the anther and could play a crucial role in fertility restoration. Similar results showed that the restorer genes in CMS functioned in pollen during restoration in other crops, such as rice, although their molecular regulative mechanisms were different [52,53,54]. At the temporal expression level, they peaked at bud lengths of 4, 2, and 4 mm in R2000, CLR650, and Zhehuhong, respectively, all of which were in the early stage of bud development. It was similarly reported that the process of microspore formation and meiosis in strawberry was mainly concentrated 18–14 days before flowering (bud length was 1–3 mm) [55]. In addition, in diploid yellow potato (*Solanum tuberosum* L. Phureja Group), the meiosis was completed when the bud length was 7.5–8.9 mm [56]. Therefore, in this study, higher expression levels of restorer genes can be detected in the early stage of bud formation.

## 4. Materials and Methods

### 4.1. Plant Materials and Crop Husbandry

The plant materials were three restorer lines R2000 (R), CLR650 (C), and Zhehuhong (Z). R2000 was purchased from INRA (Paris, France), CLR650 was a gift from Prof. Weijiang Chen (Hunan Academy of Agricultural Sciences), and Zhehuhong was provided by our group [32]. Zhehuhong was confirmed by test-crossing between R and the Ogura CMS sterile line HA3 (S). F_1_ was 100% fertile; however, the petal color was not red, but orange (Figure 6). The red petal of Zhehuhong is a unique characteristic of restorers, and the gene controlling the red petal was located in chromosome A7.

The experiment was carried out at the experimental station of the Zhejiang Academy of Agricultural Sciences on 16 October 2021 and 21 October 2022. The planting method was direct seeding, with about 5 seeds in each hole. A slow-release compound fertilizer (N: P_2_O_5_: K_2_O = 15: 7: 8) with 15 kg ha^−1^ of boron (B) was applied to the soil as the base fertilizer, at the amount of 750 kg ha^−1^. The slow-release compound fertilizer was purchased from the Hubei Yishizhuang Agricultural Science and Technology Company, Limited (Yichang, China). When plants entered the bud stage, a topdressing was applied with urea, at the amount of 75 kg ha^−1^. The plants were not irrigated during the growth period. The temperature and precipitation are shown in Figure 7. Overall, the trend of temperature was the same in two years, but the lowest temperature appeared in different months. The rainfall in 2021–2022 was clearly higher than that from January onwards, particularly in March (Figure 7).

### 4.2. Experimental Design

The experiment had a completely randomized block design. Three restorer lines, R2000, CLR650, and Zhehuhong, served as the treatment, using R2000 as the control. The experiment was conducted in three replications. The plot area was 20 m^2^, with a planting density of 150,000 plants ha^−1^.

### 4.3. Sampling

The leaves from seedlings of R2000, CLR650, and Zhehuhong were sampled and stored at −80 °C for the cloning, sequencing, and molecular marker detection of the restorer gene *PPRB*. In order to study the spatial expression level of *PPRB*, samples of the leaves, root tips, young stems, bracts, petals, anthers, and stigmas of the three restorer lines, from the early flowering stage to the full flowering stage, were taken for the measurement of expression levels. Samples of buds of 2, 3, 4, 5, and 7 mm in length were taken to study the temporal expression differences of *PPRB* (Figure 8).

### 4.4. Mitosis Analysis

The analysis of mitosis in the three restorer lines was performed according to the method reported by Wang et al. (2021) [57]. Briefly, seeds of R2000, CLR650, and Zhehuhong were soaked in ddH_2_O for 6 h. The seeds then were transferred to a Petri dish covered with sterilized filter paper and germinated under dark conditions at 25 °C. During the process of germination, the filter paper remained moist. After seeds germinated and the root tip grew to a length of 2 cm, the root tips were cut and put into 0.002 mol L^−1^ of 8-hydroxyquinoline solution. The root tips in solution were kept under dark conditions at room temperature for 2 h. The root tips were transferred to Carnoy’s fixative solution for 24 h. The fixed root tips were stored in 70% ethanol solution at 4 °C.

The root tips were washed three times with distilled water for 5 min each time. The washed root tips were transferred into 12 mol L^−1^ of HCl solution in a 60 °C water bath for 6 min. Then, the samples were washed in distilled water three times again. The treated root tips were transferred to a clean glass slide, cut into small pieces, and crushed by sterilized tweezers. After removing large pieces of residue, one drop of carbol fuchsin staining solution was added and the root tips were stained for 10 min. The sample was covered by a clean glass cover. Then, 45% acetic acid solution was dripped from one side of the cover glass and the excess solution was removed with absorbent paper at the other side. Finally, the sample was placed under a fluorescence inverted microscope (Zeiss Axio Vert. A1, Oberkochen, Germany). Lastly, photos of the chromosomes were taken and the number of chromosomes was counted when somatic cells with a clear division phase were observed.

### 4.5. Meiosis Analysis

The analysis was conducted according to the method described by Zhang et al. (2013) [58]. In the budding stage, young buds, 1–2 mm in length, of R2000, CLR650, and Zhehuhong were selected and placed in Carnoy’s fixative solution. They were replaced 3–4 times with fresh Carnoy’s fixative solution every 2 h until the Carnoy’s fixative solution no longer showed the clear yellow color. The samples were taken out from the fixative solution and rinsed several times with distilled water, until the odor of acetic acid completely dissipated. The buds were peeled with sterilized tweezers and placed on a clean glass slide. After removing large pieces of residue, one drop of carbol fuchsin staining solution was added to the sample and stained for 15 min. After covering with the cover glass, filter paper was used to cover the glass and the excess staining solution was removed by finger pressure. Finally, the cover glass was vertically pressed with a thumb and the samples were observed under a fluorescence inverted microscope. The clear meiosis phases of R2000, CLR650, and Zhehuhong were recorded.

### 4.6. Molecular Marker Analysis of Exogenous Radish Fragment

According to the published patent [33], a total of 79 molecular markers were chosen for the analysis of exogenous radish fragment length. These markers were divided into five regions. Ten molecular markers of RMA01–RMA10 were included in the first grou; 14 molecular markers of RMB01–RMB12, E35M62, and OPF10 were in the second, with the restorer gene *Rfo* also located in this group. A total of 34 molecular markers of RMC01–RMC33, E38M60, and OPC2 were located in the third; nine molecular markers of E33M47, E32M50, OPN20, OPH15, IN6RS4, E33M58, E32M9A, E32M59B, and OPH03 were in the fourth, with the genes controlling glucosinolate content also located in this group, and 11 molecular markers of RME01–RME10 and IN10RS4 were in the fifth. The total DNA of R2000, CLR650, and Zhehuhong was used as a template for PCR amplification. The amplified products were separated by electrophoresis in 1.0% agarose gel. Finally, the amplified bands were photographed, recorded, and sorted. The sequences of primers are listed in Appendix A.

### 4.7. Cloning and Sequencing of Restorer Gene PPRB

The total DNA of R2000, CLR650, and Zhehuhong was extracted according to the instructions of the SimGen Plant DNA Kit (Hangzhou Simgen Biotechnology CO. Ltd., Hangzhou, China). The restorer gene *Rfo (PPRB*) sequence was obtained from the National Center for Biotechnology Information (>AJ550021.2:88044-90235 *Raphanus sativus D81Rfo* (restoration of fertility) genetic region). The restorer gene *PPRB* was divided into three segments for amplification. Primer sequences are listed in Appendix A. The amplification products were sent to Shanghai Sangon Biotechnology Co., Ltd. (Shanghai, China) for sequencing and analysis of the result.

### 4.8. Spatiotemporal Expression Difference Analysis of Restorer Gene PPRB

The leaves, root tips, young stems, bracts, petals, anthers, stigmas, and different lengths of buds of R2000, CLR650, and Zhehuhong were used for RNA extraction, according to the instructions of the QIAGE kit (4016050, Hangzhou Simgen Biotechnology Co., Ltd., Hangzhou, China). The total RNA concentration was measured using a Thermo ultramicro spectrophotometer (NanoDro 2000C, Shanghai Bajiu Industrial Co., Ltd., Shanghai, China). cDNA was transcribed using PrimerScript^TM^ 1st Strand cDNA synthesis (Perfect Real Time) (6110A, Takara, TaKaRa Biotechnology (Dalian) Co., Ltd., Kusatsu, Japan), according to the instructions. The quantification of gene expression amount was performed on a Roche LightCyle 480 fluorescence quantitative PCR instrument (Roche, Basel, Switzerland), using the TB Green Primix Ex Taq ^TM^ II (Tli RNaseH Plus) and ROX plus enzyme kit (Takara, Kyoto, Japan). Rapeseed *ACTIN* was used as the reference gene, and the sequences of the primer used in this study are listed in Appendix A.

### 4.9. Statistics

SPSS statistical software (version26.0, Chicago, IL, USA) was used for analysis of variance (ANOVA). The significant differences of gene expression amount were analyzed (*p* < 0.05). The transcript levels of PPRB were calculated using 2^−ΔΔCt^ [59].

## 5. Conclusions

In the current study, a new Ogura CMS restorer Zhehuhong was successfully bred by distant hybridization between rapeseed and red petal radish. The cytological and molecular evidence indicated that Zhehuhong is a stable restorer with the number of chromosomes (2n = 38), and a new restorer with significant SNPs in the restorer gene (*PPRB*), as compared with R2000. The length of exogenous radish fragments in Zhehuhong was different from the lengths of R2000 and CLR 650 according to molecular marker detection analysis. *PPRB* had the highest expression level in the anthers of R2000, CLR650, and Zhehuhong, and at a 2 mm bud in Zhehuhong. This study provided molecular evidence for the unique characteristics of the newly bred restorer line, Zhehuhong. However, the excessive exogenous radish chromosome fragment should be further improved by shortening it and removing the unnecessary adverse genes controlling the glucosinolate content.

## Figures and Tables

**Figure 1 plants-13-01703-f001:**
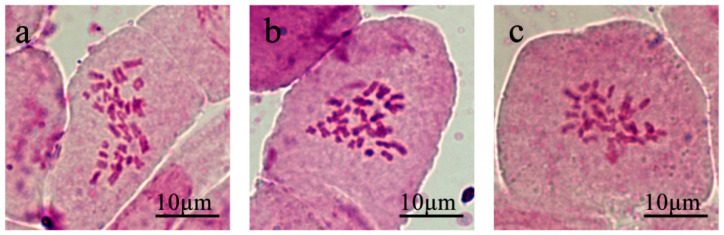
Mitotic observation of R2000, CLR650, and Zhehuhong: (**a**) R2000, (**b**) CLR650, (**c**) Zhehuhong.

**Figure 2 plants-13-01703-f002:**
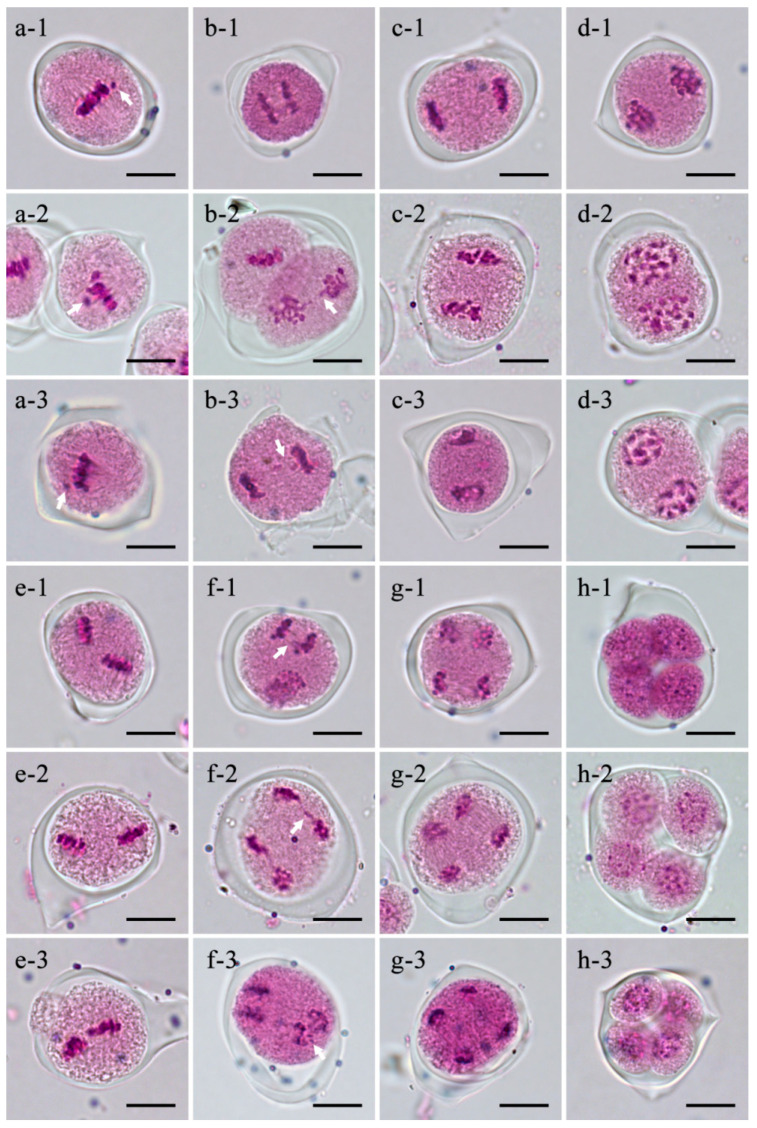
Meiosis observation of R2000, CLR650, and Zhehuhong: (**a-1**–**a-3**) meiotic metaphase I, (**b-1**–**b-3**) meiotic anaphase I, (**c-1**–**c-3**) meiotic telophase I, (**d-1**–**d-3**) meiotic prophase II, (**e-1**–**e-3**) meiotic metaphase II, (**f-1**–**f-3**) meiotic anaphase II, (**g-1**–**g-3**) meiotic telophase II, and (**h-1**–**h-3**) tetrad stage of R2000, CLR650, and Zhehuhong, respectively. The white arrows indicate chromosomal bridges. Bar = 10 μm.

**Figure 3 plants-13-01703-f003:**
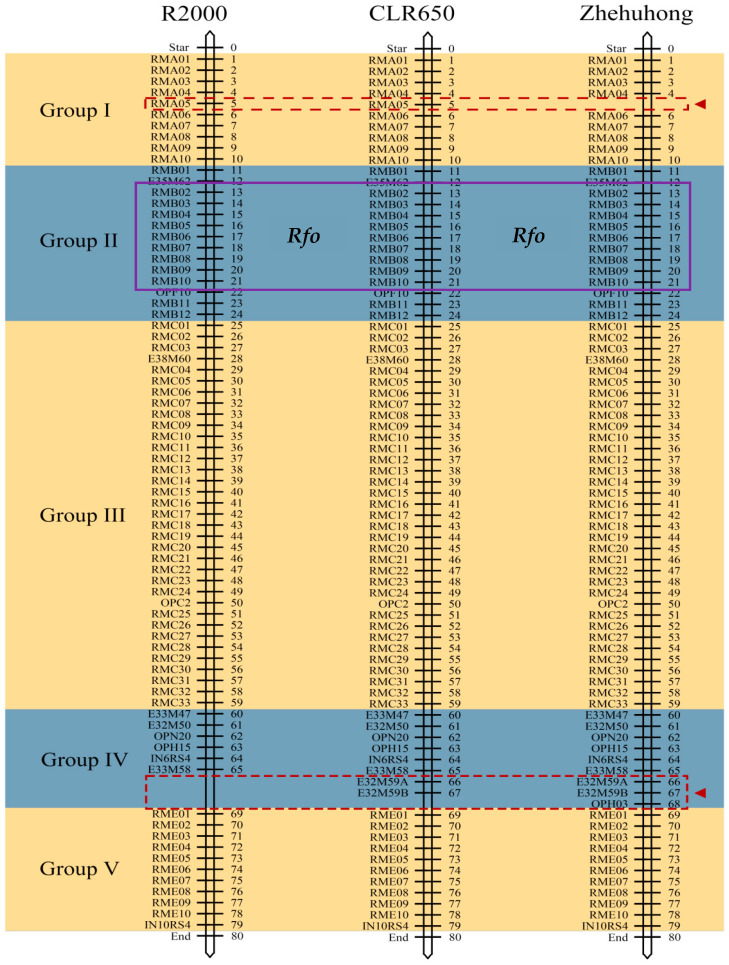
Amplification results of radish-specific markers in R2000, CLR650, and Zhehuhong. The arrows indicate the differential site in three restorer lines. The blue areas represent different regions of selected markers. The purple box indicates the *Rfo* (*PPRB*) located in this region [33].

**Figure 4 plants-13-01703-f004:**
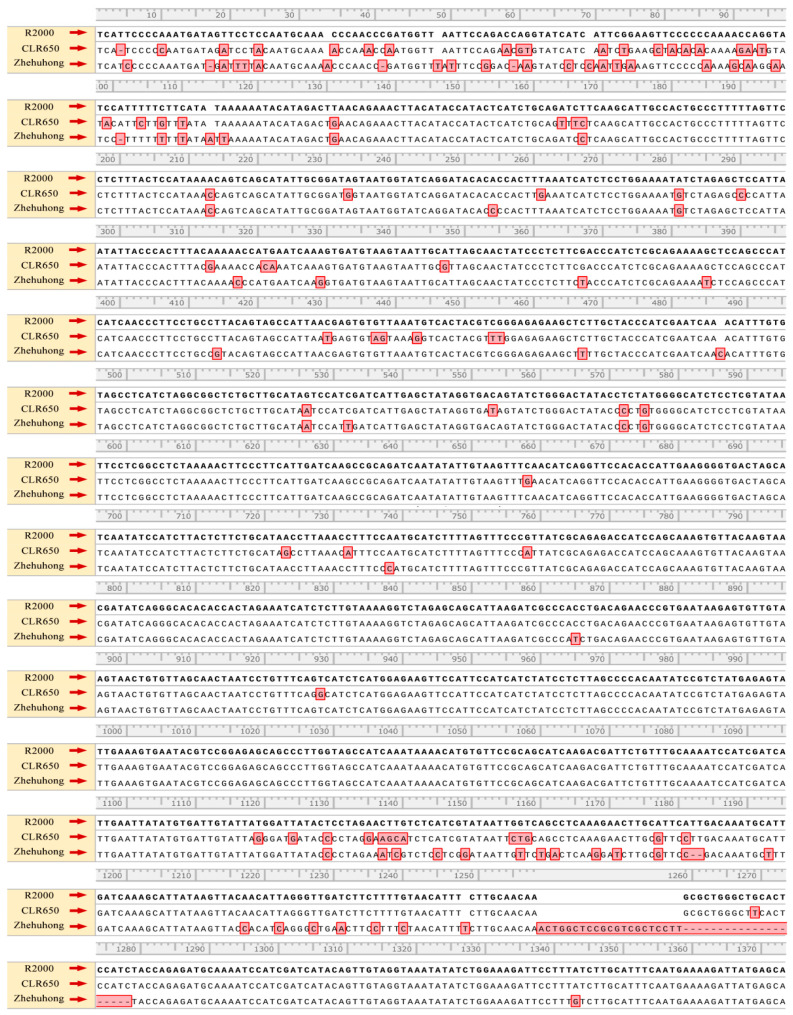
Sequencing results of amplification product of restorer gene *PPRB* in R2000, CLR650, and Zhehuhong. Red boxes represent single nucleotide polymorphism (SNP) and indels.

**Figure 5 plants-13-01703-f005:**
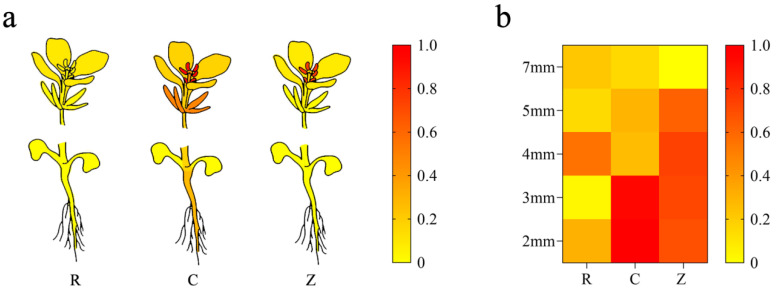
(**a**) Spatial and (**b**) temporal expression of restorer gene *PPRB* in different organs in R2000, CLR650, and Zhehuhong. The red and yellow colors represent the up- and down-regulated expression levels of *PPRB*.

**Figure 6 plants-13-01703-f006:**
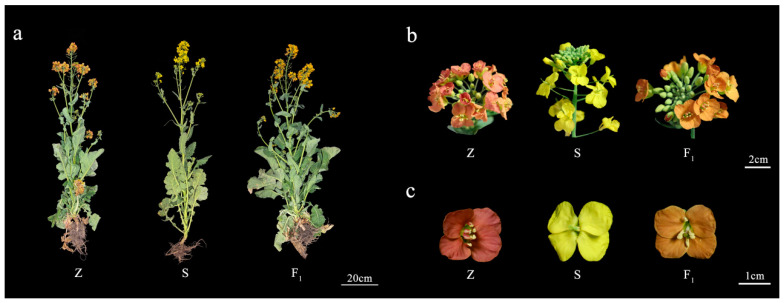
Plant morphology and flower organ morphology of Zhehuhong, Ogura CMS sterile line HA3, and their hybrid F_1_. (**a**–**c**), Plant morphology, inflorescence, and petal morphology of Zhehuhong, HA3, and their hybrid F_1_, respectively. Z: Zhehuhong, S: Ogura CMS sterile line HA3, F_1_: the hybrid offspring of Zhehuhong (Z, male parent) and HA3 plant (S, female parent).

**Figure 7 plants-13-01703-f007:**
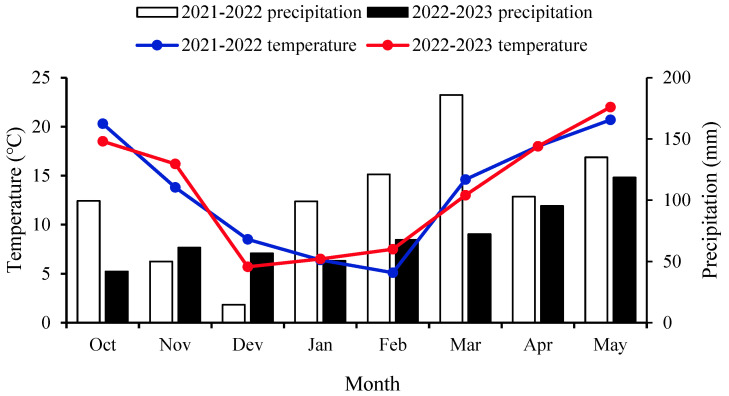
Temperature and precipitation during rapeseed growth in 2021–2022 and 2022–2023.

**Figure 8 plants-13-01703-f008:**
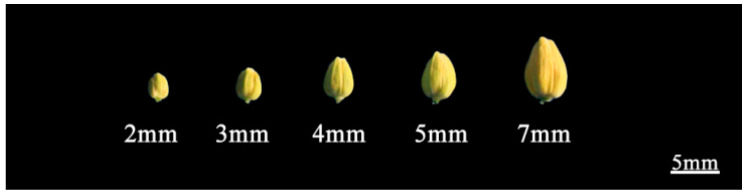
Different sizes of buds for quantitative real-time PCR analysis.

## Data Availability

The data in this study can be made available under reasonable request from the corresponding author.

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
