# Peer review of "Cytological and Molecular Characterization of a New Ogura Cytoplasmic Male Sterility Restorer of Brassica napus L."

_plants, 2024, doi:10.3390/plants13121703_

Round 1

Reviewer 1 Report

Comments and Suggestions for Authors

Thank you for considering me to review the manuscript titled: Cytological and Molecular Characterization of a New Ogura CMS Restorer of Brassca napus L. The manuscript provides a reasonably robust dataset on providing New cytoplasmic male sterile Restorer of Brassca napus. With some minor revisions, the manuscript appears suitable for acceptance after minor revisions.

Suggestions:

Ensure thorough English language editing throughout the manuscript to rectify grammatical errors and shorten lengthy sentences.

Please enhance the abstract by providing more descriptive details regarding methodology and emphasizing the results obtained. Abbreviations should be provided at the first appearance as CMS (cytoplasmic male sterile).

The short introduction should be extended and improved by elaborating on the importance of cytoplasmic male sterility in improving rapeseed. Highlight the efficacy of CMS Restorer in enhancing rapeseed production. Provide the contribution of the cytoplasmic male sterility and CMS hybrid system in sustainable rapeseed production. Moreover, it clarifies the knowledge gap and hypothesis and extends the objectives.

The results section is well presented and structured.

The discussion section should be improved, it should benefit from incorporating recent literature to support statements and clarify ambiguous points.

The material and method section is well structured.

Ensure consistent use of journal abbreviations and proper citation formatting throughout the manuscript. Some journals are abbreviated as " Int J Mol Sci." line 415, J. Agric. Sci” line 417,….. while others are not abbreviated as “Plant Biotechnology Journal” line 420, “Theoretical and Applied Genetics” line 423…..

Comments on the Quality of English Language

Moderate editing of English language required

Author Response

Reponses to Reviewer 1

Thank you for considering me to review the manuscript titled: Cytological and Molecular Characterization of a New Ogura CMS Restorer of Brassca napus L. The manuscript provides a reasonably robust dataset on providing New cytoplasmic male sterile Restorer of Brassca napus. With some minor revisions, the manuscript appears suitable for acceptance after minor revisions.

R: very thank you for your positive comments. All revisions can be clearly found in green background. Other revisions were also made in grey background.

Suggestions:

Ensure thorough English language editing throughout the manuscript to rectify grammatical errors and shorten lengthy sentences.

R: Thank you for your comments. We used the agency recommended by the journal to correct English. I also attach the certification provided by the agency.

Please enhance the abstract by providing more descriptive details regarding methodology and emphasizing the results obtained. Abbreviations should be provided at the first appearance as CMS (cytoplasmic male sterile).

R: Thank you. We have re-structured this section as suggested. Abbreviations are provided as well.

The short introduction should be extended and improved by elaborating on the importance of cytoplasmic male sterility in improving rapeseed. Highlight the efficacy of CMS Restorer in enhancing rapeseed production. Provide the contribution of the cytoplasmic male sterility and CMS hybrid system in sustainable rapeseed production. Moreover, it clarifies the knowledge gap and hypothesis and extends the objectives.

R: Thank you for your suggestion. We have added information as suggested and revised the section of the knowledge gap and hypothesis. The objectives were also extended. 

The results section is well presented and structured.

R: Thank you. 

The discussion section should be improved, it should benefit from incorporating recent literature to support statements and clarify ambiguous points.

R: Thank you for your suggestion. We have provided more information on result of the recent studies. The ambiguous points were also thoroughly revised.

The material and method section is well structured.

R: Thank you.

Ensure consistent use of journal abbreviations and proper citation formatting throughout the manuscript. Some journals are abbreviated as " Int J Mol Sci." line 415, J. Agric. Sci” line 417,….. while others are not abbreviated as “Plant Biotechnology Journal” line 420, “Theoretical and Applied Genetics” line 423…..

R: Thank you for your careful examination. We have corrected the formatting of literatures.

Reviewer 2 Report

Comments and Suggestions for Authors

The manuscript submitted for revision (plants-3041040) clearly presents a very important research for heterosis utilization in rapeseed to increase the seed yield. In this study , a new Ogura CMS restorer Zhehuhong line was successfully created by distatnt hybridization between rapeseed and red petal radish. This line can be used effectively in heterosis breeding to restore F1 generation fertility in Brassica napus L. I have noticed in the text some minor mistakes which need a correction , as follows: line 38 - should be France (name of the country) not French; line 45 - the word in is doubled; line 127 - A marker ( the space is needed); line 390 - 2 mm of bud (not bus). After the minor correction I recommend this paper to publish in Plants. 

Comments on the Quality of English Language

Quality of English language is good. See comments above. 

Author Response

Response to Reviewer 2

The manuscript submitted for revision (plants-3041040) clearly presents a very important research for heterosis utilization in rapeseed to increase the seed yield. In this study , a new Ogura CMS restorer Zhehuhong line was successfully created by distatnt hybridization between rapeseed and red petal radish. This line can be used effectively in heterosis breeding to restore F1 generation fertility in Brassica napus L. I have noticed in the text some minor mistakes which need a correction , as follows: line 38 - should be France (name of the country) not French; line 45 - the word in is doubled; line 127 - A marker ( the space is needed); line 390 - 2 mm of bud (not bus). After the minor correction I recommend this paper to publish in Plants. 

R: Very thank you for your suggestions. We have corrected those mistakes as you pointed out. Furthermore, we have corrected all possible mistakes throughout the manuscript. All revisions can be found in blue background. And other corrections in grey background.

Reviewer 3 Report

Comments and Suggestions for Authors

The article "Cytological and Molecular Characterization of a New Ogura CMS Restorer of Brassica napus L."reports the results of a systematic study on the development of a new restorer Zhehuhong for the Ogura CMS line of Brassica napus capable of restoring 100% fertility in the brassica hybrids as compared to the previously developed restorer lines R2000 and CLR650. The spelling of brassica in the title should be corrected. The mitotic figures clearly showing the chromosomes should be given. The radish lines used for the development of the three restorer lines should be given as the sequence data with different SNPs and Indels indicates that different radish lines must have been used to develop these restorer lines having the same effective restorer gene PPRB. The mitotic and meiotic cytological data indicates that the restorer lines with different chromosome numbers than 2n=38 and laggards may not be stable lines for use in hybrid breeding programs. The introgression of the radish chromosome segment carrying the restorer gene PPBR on a particular brassica genome, chromosome and its size variation in all the three restorer lines should be given. The source of 79 molecular markers and their recombinant frequencies in cM should be given clearly indicating the location of the PPBR genes in one of the five categories of the molecular markers. The gene for red petal colour associated with the Zhehuhong indicating a unique source different from those of the other sources should also be indicated. There are numerous grammatical mistakes and wrong use of the English scientific words used in the article which have been highlighted in the revised manuscript attached. These should be substituted and corrected if decide to revise the manuscript. All the sections of the M&M should be written in the past tense.

Comments on the Quality of English Language

These have  been already indicated in the comments for the authors above.

Author Response

Response to Reviewer 3

The article "Cytological and Molecular Characterization of a New Ogura CMS Restorer of Brassica napus L."reports the results of a systematic study on the development of a new restorer Zhehuhong for the Ogura CMS line of Brassica napus capable of restoring 100% fertility in the brassica hybrids as compared to the previously developed restorer lines R2000 and CLR650. The spelling of brassica in the title should be corrected.

R: Thank you for your careful review. We have corrected. All revisions were made in yellow background and others were in grey background.

The mitotic figures clearly showing the chromosomes should be given.

R: Heartfully speaking, we also do not satisfy these figures because of low resolution. However, we tried our best to improve it. The figures in the manuscript are the best due to the limitation of our technology. I really hope you can forgive us that we cannot provide better quality figures. Thank you for your understanding again.

The radish lines used for the development of the three restorer lines should be given as the sequence data with different SNPs and Indels indicates that different radish lines must have been used to develop these restorer lines having the same effective restorer gene PPRB.

R: Thank you. We have added the information L217~L222.

The mitotic and meiotic cytological data indicates that the restorer lines with different chromosome numbers than 2n=38 and laggards may not be stable lines for use in hybrid breeding programs.

R: Thank you for your comments. You are right, it is not so stable in hybrid breeding programs. For R2000, which is widely applied in many countries, still have some sterile plants in hybrids. For CLR650 and Zhehuhong, the longer exogenous radish fragment might have greater effects on the hybrids. Therefore, further improvement of those restorers are required.

The introgression of the radish chromosome segment carrying the restorer gene PPBR on a particular brassica genome, chromosome and its size variation in all the three restorer lines should be given.

R: Thank you. We have added the information L256~L262.

The source of 79 molecular markers and their recombinant frequencies in cM should be given clearly indicating the location of the PPBR genes in one of the five categories of the molecular markers.

R: Thank you for your suggestion. We have added the location of PPRB in the Figure 3. However, the exact location is unknown. The source of 79 molecular markers was from other researchers, they also did not provide this information for their recombinant frequencies in cM. It is difficult now to provide these information at current stage. Hope you can understand us. Thank you.

The gene for red petal colour associated with the Zhehuhong indicating a unique source different from those of the other sources should also be indicated.

R: Thank you. We have added this information L307~L308.

There are numerous grammatical mistakes and wrong use of the English scientific words used in the article which have been highlighted in the revised manuscript attached. These should be substituted and corrected if decide to revise the manuscript. All the sections of the M&M should be written in the past tense.

R: Thank you for your suggestion and kind help to improve the quality of our manuscript. We have thoroughly revised the manuscript as you pointed in the attached file. 
